# RNA sequencing provides insights into the evolution of lettuce and the regulation of flavonoid biosynthesis

Lei Zhang[1], Wenqing Su[1], Rong Tao[1], Weiyi Zhang[1], Jiongjiong Chen[1], Peiyao Wu[1], Chenghuan Yan[1], Yue Jia[1], Robert M. Larkin[1], Dean Lavelle[2], Maria-Jose Truco[2], Sebastian Reyes Chin-Wo [2], Richard W. Michelmore[2] & Hanhui Kuang[1]

Different horticultural types of lettuce exhibit tremendous morphological variation. However, the molecular basis for domestication and divergence among the different horticultural types of lettuce remains unknown. Here, we report the RNA sequencing of 240 lettuce accessions sampled from the major horticultural types and wild relatives, generating 1.1 million single-nucleotide polymorphisms (SNPs). Demographic modeling indicates that there was a single domestication event for lettuce. We identify a list of regions as putative selective sweeps that occurred during domestication and divergence, respectively. Genome-wide association studies (GWAS) identify 5311 expression quantitative trait loci (eQTL) regulating the expression of 4105 genes, including nine eQTLs regulating genes associated with flavonoid biosynthesis. GWAS for leaf color detects six candidate loci responsible for the variation of anthocyanins in lettuce leaves. Our study provides a comprehensive understanding of the domestication and the accumulation of anthocyanins in lettuce, and will facilitate the breeding of cultivars with improved nutritional value.

[1] Key Laboratory of Horticultural Plant Biology, Ministry of Education, Key Laboratory of Horticultural Crop Biology and Genetic improvement (Central Region), MOA, College of Horticulture and Forestry Sciences, Huazhong Agricultural University, Wuhan 430070, People's Republic of China. [2] Genome Center and Department of Plant Sciences, University of California, Davis, CA 95616, USA. Correspondence and requests for materials should be addressed to H.K. (email: kuangfile@mail.hzau.edu.cn)

Lettuce (*Lactuca sativa*) is one of the most important vegetables worldwide. It is a member of the Compositae (Asteraceae) family which contains an exceedingly large number of species of flowering plants[1]. The wild ancestor of lettuce is believed to be *Lactuca serriola*[2]. Lettuce is more distantly related but sexually compatible with *Lactuca saligna* and *Lactuca virosa*[3]. Cultivated lettuce was first recorded on the walls of Egyptian tombs ca. 2500 BCE, indicating that lettuce has been cultivated for at least 4500 years[3,4]. Based on their morphological characteristics, cultivars of lettuce are classified into six horticultural types: butterhead, crisphead, looseleaf, romaine, stem, and oilseed. The former four types are harvested and consumed for their leaves and they are collectively referred to as "leafy" types. In contrast, stems of stem lettuce (mainly cultivated in Asia) are harvested and consumed, while seeds from some cultivars are harvested for oil production in Egypt[4]. Leafy lettuce appeared in China at least 1700 years ago during the Jin dynasty, as recorded by an ancient Chinese book "Zhou Hou Bei Ji Fang" (A Handbook of Prescriptions for Emergencies, 340 CE). Leafy lettuce was then developed into stem lettuce, which has thick succulent stem and reduced leaves. Stem lettuce was first reported ~900 years ago, as recorded by another ancient Chinese book "Dongjing Meng Hua Lu" (The Eastern Capital: A Dream of Splendor, 1127 CE).

During the initial domestication and subsequent improvement (diversification) of lettuce, plant shape (including leaf and stem shapes) has changed dramatically. Besides the variations among different horticultural types, cultivars within a type may also exhibit considerable phenotypic variations, such as color variation. Most lettuce cultivars have green leaves, but others may have red leaves. The red color is caused by the accumulation of flavonoids, which contribute to both plant fitness and to the nutrition of consumers[5]. However, genetic studies underlying the phenotypic variation of lettuce are limited[6].

The development of next-generation sequencing (NGS) technology has revolutionized life science research, especially genetics and evolution. NGS has been used to study the evolution and domestication of important crops, such as cucumber[7], maize[8,9], rice[10], soybean[11], and tomato[12]. NGS also makes linkage or association analyses possible for most (if not all) species, leading to the identification of many candidate genes responsible for important traits in crops, such as grain size[13], drought tolerance[14], and kernel oil[15]. However, the domestication of lettuce, its population structure, and the genetic and molecular mechanisms underlying the variations among different horticultural types remain to be investigated[16].

In this study, 240 wild and cultivated lettuce accessions were studied comprehensively using RNA-Seq. The origin and domestication of lettuce were analyzed using a large number of SNPs obtained for wild and cultivated lettuce. The population structure of cultivated lettuce and the relationships among different horticultural types were studied in detail. Artificial selection during domestication and subsequent cultivar differentiation were also analyzed. The SNPs were used to perform a GWAS analysis of leaf color in lettuce. Furthermore, the SNP and expression data were combined to investigate eQTLs for genes involved in anthocyanin biosynthesis. The results in this study not only shed light on lettuce evolution but also provide useful information for future lettuce breeding programs.

## Results

**RNA sequencing and SNP identification**. A total of 240 accessions of *Lactuca* spp., which represented most of the phenotypic diversity in our collection, were selected for genome-wide analysis (Supplementary Fig. 1, Supplementary Data 1). This set of germplasm includes 31 accessions of wild lettuce (24 *L. serriola*, 3 *L. saligna*, and 4 *L. virosa*) and 6 intermediate accessions that have both wild and cultivated characteristics. The other 163 accessions are cultivars, including 28 butterhead, 19 crisphead, 17 looseleaf, 3 oilseed lettuce, 31 romaine, and 24 stem lettuce. Forty-one cultivars are atypical of any horticultural type and are referred to as "atypical type" hereafter. In addition to these 200 accessions, we also included 40 inbred lines derived from a cross between a looseleaf cultivar (S1) and a stem lettuce cultivar (Y37), which were only used for eQTL and GWAS analysis in this study. RNA was extracted from the young leaves of ~2.5-month-old plants from each accession. RNA sequencing using Illumina technology generated 4.51 billion paired-end reads of 125 bp (1.13 Tb of sequences), after filtering out the low-quality reads. On average, 18.8 million paired-end reads (4.70 Gb) were obtained for each sample. The filtered reads from each accession were mapped to the *L. sativa* cv. Salinas genome[17]. The mapping rate varied from 91.08 to 98.54% among different accessions, with an average of 97.28%.

Using the mapping results, 1,133,865 high-quality SNPs were detected using a series of filtering approaches. More than half of them (712,650 or 62.85%) are unique to *L. saligna* and *L. virosa* relative to the *L. serriola* and *L. sativa* groups (Supplementary Fig. 2). Sixteen accessions were excluded from further analyses because they had excessive heterozygosity (Supplementary Data 1).

As expected from RNA-Seq data, the majority of SNPs (94.64% or 1,073,169) were located within genes (Supplementary Fig. 3, Supplementary Table 1) and the SNP density was consistent with the density of genes (Supplementary Fig. 4). The potential effects of SNPs on genes were then investigated and a total of 2346 SNPs (large-effect SNPs) in 2035 genes may have a major impact on gene function due to premature stop codon, induced disruptive splice variants, etc. (Supplementary Note 1, Supplementary Data 2 and Supplementary Table 1–3).

SNP loci might have missing data for some accessions due to low expression of some genes. For further studies, the missing data were imputed. A total of 344,222 SNPs with missing rates ≤0.8 were filled using fillGenotype[18] (Supplementary Note 2, Supplementary Fig. 5).

**A single domestication for lettuce**. Using accession "W17" (*L. virosa*) as an out-group, a maximum-likelihood phylogenetic tree was constructed to show the phylogenetic relationships among wild and cultivated lettuce (Fig. 1a). In the tree, all cultivars form a monophyletic clade, which is clearly separated from the wild accessions. This result indicates that cultivated lettuce probably originated from a common ancestor (i.e., a single domestication event). Among the wild species, *L. serriola* is closer to cultivated lettuce than either *L. saligna* or *L. virosa*, supporting the hypothesis that *L. serriola* was the progenitor of cultivated lettuce[2]. The six intermediate accessions were located between *L. serriola* and cultivated lettuce in the phylogenetic tree. Cultivated accessions formed several subgroups representing each horticultural type (Fig. 1a, Supplementary Fig. 6). The only exception was the looseleaf horticultural type. Cultivars of the looseleaf type were distributed irregularly across the *L. sativa* clade. This distribution is consistent with the high phenotypic variation of looseleaf cultivars. The accessions that did not have distinctive characteristics of any horticultural type (atypical type) were also distributed irregularly across the *L. sativa* clade.

The phylogenetic relationships of the different lettuce groups were also supported by principal component analysis (PCA) (Fig. 1b). The PCA plot indicates that cultivated lettuce is more closely related to *L. serriola* than either *L. saligna* or *L. virosa*.

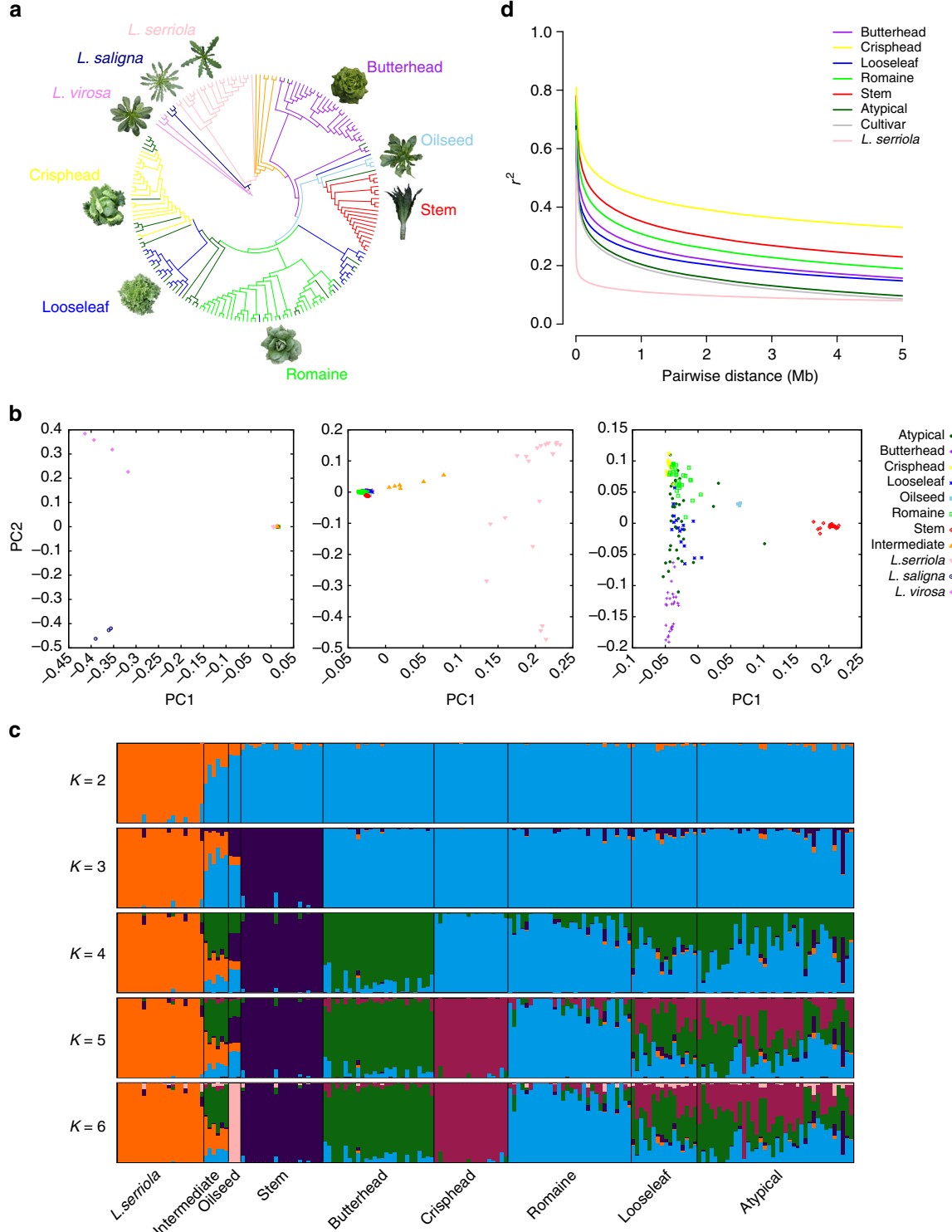

**Fig. 1** Population genetic analyses of wild and cultivated lettuce. **a** Maximum-likelihood phylogenetic tree constructed using 503,369 SNPs. Colors indicate the following groups: violet, *L. virosa*; navy blue, *L. saligna*; pink, *L. serriola*; orange, intermediate group; dark green, atypical group; sky blue, oilseed; red, stem; green, romaine; blue, looseleaf; yellow, crisphead; purple, butterhead. **b** PCA of *Lactuca* accessions. The first two principal components were used to visualize the relationship among individuals and groups. Each point represents an independent accession of *Lactuca*. From left to right are: the results of PCA analysis on all *Lactuca* accessions; the results of PCA analysis with the exclusion of *L. virosa* and *L. saligna*; the results of PCA analysis with the exclusion of the intermediate type, *L. serriola*, *L. virosa* and *L. saligna*. **c** Population structure analysis with different numbers of clusters (*K* = 2–6). The *y* axis quantifies cluster membership and the *x* axis shows the different populations. **d** LD decay of *L. sativa* and *L. serriola* measured by $r^2$

When these two distantly related wild species were excluded, the cultivated varieties grouped together and were clearly separated from *L. serriola*, consistent with our conclusion that there was a single domestication of cultivated lettuce. When *L. serriola* and the intermediate accessions were excluded, stem lettuce was clearly separated from the other horticultural types. This finding is consistent with a long history of separation between leafy lettuce and stem lettuce.

**Each horticultural type forms a distinct cluster**. To further investigate the population structure of lettuce, the Bayesian clustering program STRUCTURE[19] was used through gradually increasing the number of clusters ($K$). Different numbers of clusters were identified as $K$ was increased (Fig. 1c), and the $\Delta K$ analysis showed that $K = 2$ fits the data set best (Supplementary Fig. 7). Wild accessions formed one cluster when $K$ varied from 2 to 6. When $K = 3$, the cultivated lettuce was subdivided into two clusters, one for stem lettuce and one for all of the other horticultural types. At $K = 4$ and 5, new clusters appeared for the butterhead and crisphead types, respectively. When $K = 6$, wild lettuce and five horticultural types (butterhead, crisphead, oilseed, romaine, and stem) were assigned to independent clusters. However, cultivars of the looseleaf and the atypical types exhibited admixture proportions, consistent with the results from our phylogenetic and PCA analyses. Based on genome-wide SNP analysis and phenotypic variation, we suggest that looseleaf cultivars may have experienced frequent gene flow from other horticultural types and that the atypical types are probably derived from crosses between different horticultural types. The pairwise population differentiation level (*F*st) between different horticultural types or groups was also calculated, indicating strong population differentiation among different horticultural types (Supplementary Table 4).

**High linkage disequilibrium in cultivated lettuce**. Linkage disequilibrium (LD), measured as the correlation coefficient ($r^2$) between SNP loci, was calculated for *L. serriola* and cultivated lettuce groups (Fig. 1d). The LD decay was measured as the physical distance at which LD dropped to half its maximum value. When all genotypes (including *L. serriola* and cultivated accessions) were analyzed, the LD decay was estimated to be 140 kb ($r^2 = 0.29$). When different types were analyzed independently, their LD decay varied dramatically. For example, the genome-wide LD was 5 kb for *L. serriola* ($r^2 = 0.20$) and 200 kb for all types of cultivars combined ($r^2 = 0.33$). When different horticultural types were calculated individually, the LD decay varied from 215 kb to 2.0 Mb. The LD decay ($r^2 = 0.40$) of the crisphead type was the largest (2.0 Mb) among all horticultural types.

**Demographic history inference for lettuce**. The demographic inference tool *fastsimcoal2*[20] was used to infer the demographic history of *L. serriola* and four horticultural types (butterhead, crisphead, romaine, and stem). The looseleaf type exhibiting admixture proportions was excluded from further study due to its strong impact on the demographic analysis[21].

To help guide the development of demographic models for lettuce, one population (*L. serriola* and four individual horticultural types), two population (*L. serriola* was combined with each horticultural type), and four population (*L. serriola* was combined with three leafy horticultural types) models were tested step by step (Supplementary Note 3, Supplementary Fig. 8–10 and Supplementary Table 5–7). Then, *L. serriola* and four horticultural types (including stem lettuce) were analyzed to test whether there was single-founder or double-founder (independent domestications of stem lettuce and the other horticultural types)

event for lettuce. Gene flow among different populations was allowed in these models. Two symmetrical migration parameters were used, one for migration between different horticultural types (migration rate among horticultural types, MRHT), and the other for migration between *L. serriola* and each of the four horticultural types (migration rate between wild and cultivated lettuce, MRWC). The result showed that single-founder model (Fig. 2a) outperformed the double-founder model (Supplementary Fig. 11) ($w_i \approx 1$, Supplementary Table 8). These results are consistent with a single domestication event for lettuce. According to the maximum-likelihood point estimates of the best fitting model, domestication of lettuce occurred 10,829 years before present (y B.P.) (95% CI = 10,391–13,005 y B.P., Supplementary Table 9). This number is close to the historical record for the beginning of human-associated plant domestication, ~12,000 y B. P. in the Middle East and the Fertile Crescent[22]. The divergence of stem lettuce from the ancestral cultivated lettuce was estimated to have occurred ~1922 y B.P. (95% CI = 1730–3036 y B.P.), similar to the time of the first record of lettuce in China ~1700 y B.P. (likely first introduced into China through Silk Road). The split time of butterhead, crisphead, and romaine from the ancestral cultivated lettuce was estimated to be around 500 y B.P., which is consistent with a previous study[4]. The migration rate between horticultural types (MRHT) and between wild and cultivated lettuce (MRWC) were 7.7 and 5.5 per 100,000 alleles, respectively.

A lettuce dispersion map was proposed based on above evidences (Fig. 2b). Cultivated lettuce was likely originated in the Fertile Crescent ~10,800 y B.P. according to our demographic analysis. It spreads to Europe over the following thousands of years. During this period, several different primitive lettuce types emerged, such as oilseed lettuce for oil production and butterhead, looseleaf or romaine lettuce as leaf vegetables. Leafy lettuce types were brought to China most likely via the Silk Road ~1922 y B.P., and continuous selection resulted in a new horticultural type, stem lettuce, ~900 years ago. The cultivated lettuce was introduced into America by conquistadors in the sixteenth century, and subsequent selection resulted in the development of modern cultivars of crisphead lettuce.

**Domestication changed nucleotide and expression diversity**. Using the SNP data, genome-wide nucleotide diversity ($\pi$) for each group/type was calculated (Supplementary Table 10). The nucleotide diversity was substantially higher in *L. serriola* ($4.84 \times 10^{-3}$) than in all cultivated lettuce combined ($2.14 \times 10^{-3}$), consistent with the genetic bottleneck that occurred during domestication. Among the five cultivated types, looseleaf type ($2.11 \times 10^{-3}$) has the highest while crisphead type ($8.94 \times 10^{-4}$) has the lowest nucleotide diversity.

The coefficient of variation (CV) of genes was calculated to assess their expression diversity for each group and type (Supplementary Table 10, Supplementary Fig. 12a). Unlike nucleotide diversity, the expression diversity of the cultivars (all types combined) was higher than *L. serriola*. Based on our finding that the expression diversity of *L. serriola* was higher than the individual horticultural types, we suggest that divergence in gene expression among different horticultural types may explain the high expression diversity of all cultivars. To investigate the selection on gene expression diversity, we compared the CV of expression between selected and non-selected genes in cultivated lettuce (see below). The CV of selected genes (51.72%) is considerably lower than that of the non-selected genes (71.26%) ($P < 2.2 \times 10^{-16}$, Student's *t* test; Supplementary Fig. 12b). This result is consistent with a previous study in maize that showed *cis*-acting regulatory variations for selected genes were removed

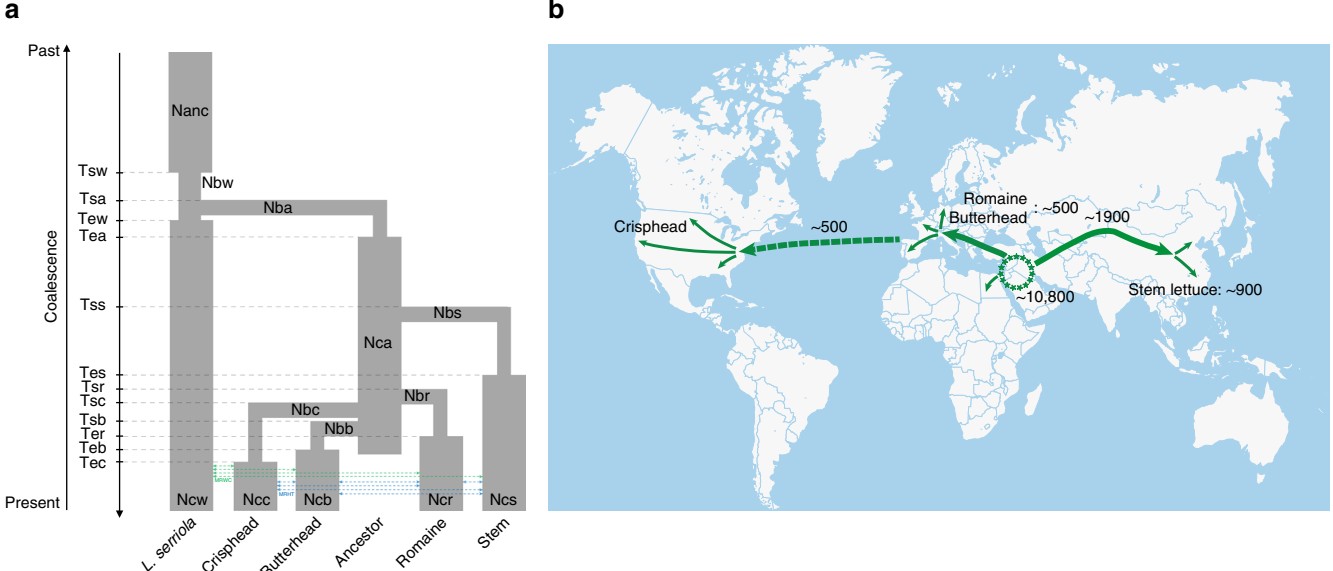

**Fig. 2** Schematic representations of the demographic history scenarios of lettuce **a** Schematic representation of demographic scenario using *Fastsimcoal2*. Gene flow among different populations was allowed. Two symmetrical migration parameters were used, one between different horticultural types (MRHT), and the other between *L. serriola* and each horticultural type (MRWC). Only the best model is shown, which assumes that there was a single domestication event and that all horticultural types originated from the same ancestral cultivated population. See Supplementary Table 9 for the meaning of the parameters and their estimated values. **b** The map showing the origin and dispersal of domesticated lettuce. The star circle indicates the center of lettuce domestication, with possible dispersal routes shown by arrows[4]. The numbers indicate the likely time (in years before present) of domestication, dispersal and origin of new horticultural types

during domestication[9]. We further compared the distribution of eQTLs between selected and non-selected genes in cultivars (Supplementary Fig. 12c–d). The percentage of selected genes (5.74%) with eQTLs was significantly lower than for non-selected genes (10.83%) ($P = 3.45 \times 10^{-16}$, Fisher's exact test). We also found less local eQTLs (54.40% in selected versus 80.83% in non-selected, $P = 4.16 \times 10^{-11}$, Fisher's exact test) and more distant eQTLs (57.60% in selected versus 34.32% in non-selected, $P = 1.41 \times 10^{-7}$, Fisher's exact test) for selected genes than for non-selected genes. Our results suggest that the low expression variation of selected genes is due to less variation of their *cis*-acting regulatory regions.

**Identification of selected and introgressed regions.** The domestication of lettuce resulted in considerable changes in many traits, including increased seed size, non-shattering, lack of spines, etc. A total of 889 candidate selective sweeps ranging from 10 to 160 kb (with an average of 41 kb in length) were detected (Fig. 3, Supplementary Data 3). These potential regions covered 1.6% (36.35 Mb) of the genome and contained 5.6% (2178 genes) of genes. GO analysis of these genes showed enrichment in stress, organ development, and metabolic processes (Supplementary Table 11). Some well-studied genes involved in these processes were identified, such as *LHY* (Late elongated hypocotyl, LG3_287450)[23], *IKU1* (Haiku, LG6_578206)[24], *PLDZ1* (Phospholipase D zeta 1, LG6_576132)[25], etc. (Supplementary Note 4). These genes might have been under selection during domestication, or were selected due to linkage with genes determining important domestication traits.

The results of previous QTL mapping were compared with selective sweeps to investigate their contribution to phenotypic changes between wild and cultivated lettuce. Previous studies identified several domestication-related QTLs in lettuce, including lobed leaf[26], seed shattering, and stem spines[16,27]. These known QTLs overlap with the putative selective sweep regions detected in our study. For example, genetic analysis of an F2

population mapped a single gene controlling lobed leaf to the interval at 116.24–118.15 Mb on LG3[26]; genetic analysis of RIL population detected a major QTL for seed shattering centered at 13.4 Mb on LG6 and stem spines centered at 302.4 Mb on LG5, respectively[16]. Several regions that have high XP-CLR values were identified in the corresponding regions. However, these regions were not grouped into a single region due to the low SNP density in this study (Fig. 3).

Different horticultural types have type-specific traits such as leaf-heading in crisphead and swollen stem in stem lettuce. A total of 172, 155, 140, and 158 regions were identified as most affected by selection, which contained 1974, 1849, 1731, and 1634 genes for butterhead, crisphead, romaine, and stem lettuce, respectively (Fig. 3, Supplementary Data 4–7). The selected genes are specific to each group, consistent with different selections leading to distinct horticultural types (Supplementary Fig. 13). Some well-studied genes associated with leaf and stem development were identified, such as *ATHB15* (*Arabidopsis thaliana* homeobox 15, LG1_164856)[28], *BOP2* (Blade on petiole 2, LG5_523032)[29], *ATH1* (*Arabidopsis thaliana* homeobox gene 1, LG3_329662)[30], and *TOAD2* (Toadstool 2, LG4_364824)[31] (Supplementary Note 5). A previous study identified several QTLs for leaf-heading in crisphead type[16], and one of them on LG3 (97.95–124.53 Mb) was coincident with the selective signals for the crisphead group (Fig. 3).

Wild *Lactuca* species have been frequently used as donors (sources) of many important traits (like resistances to diseases and pests) in modern lettuce breeding programs[32]. We identified 173 potential regions of introgression (71.88 Mb) from wild species (Fig. 3, Supplementary Fig. 14 and Supplementary Data 8–10). Most of the introgressed regions were contributed by *L. serriola* (55.9 Mb, 78%), followed by *L. saligna* and *L. virosa*. Our results are consistent with the fact that lettuce breeding has been mainly based on the utilization of *L. serriola*[32]. Seven group-specific introgressed regions were found in different horticultural types (Supplementary Table 12). For example, most of stem

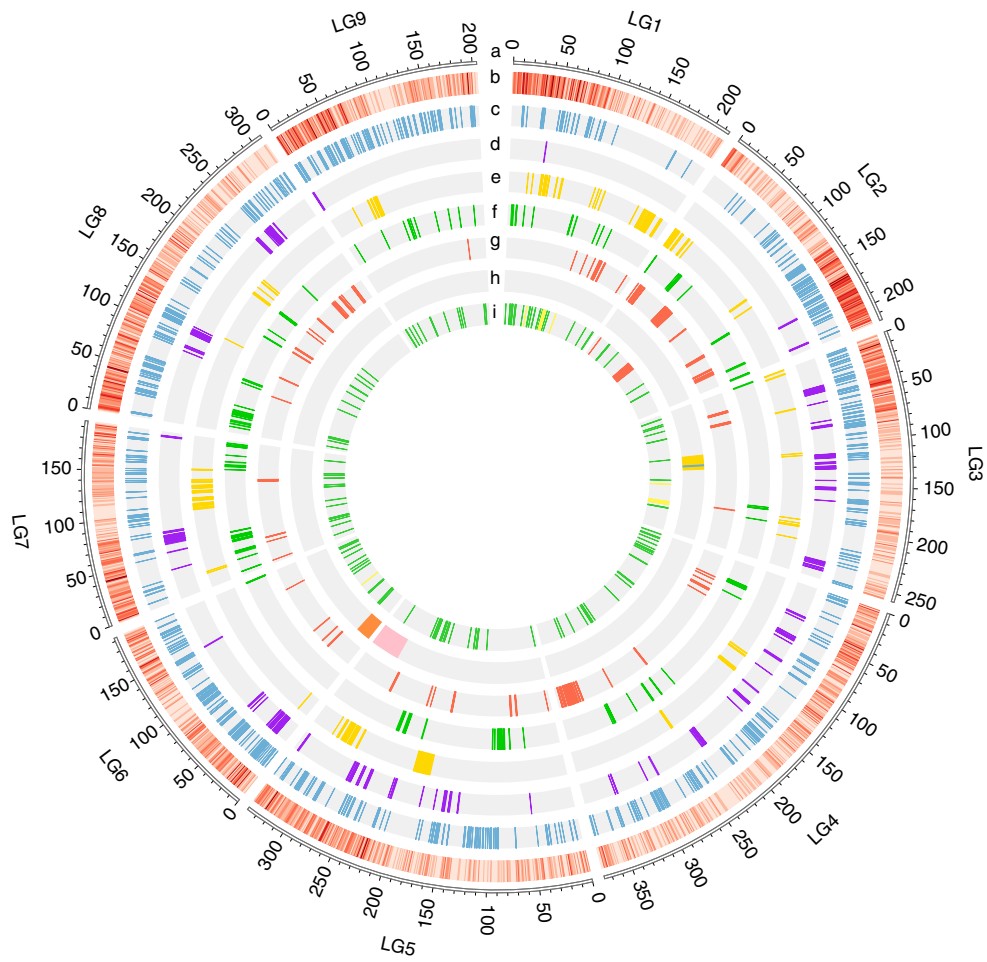

**Fig. 3** Integrated results of selection and introgression analysis. **a** Nine chromosomes of lettuce. **b** Density of identified SNP in a 1 Mb window across the genome. **c–g** Selected regions in cultivated lettuce, butterhead, crisphead, romaine, and stem. **h** QTL mapping results for leaf-heading (yellow), lobed leaf (blue), stem spines (pink), and seed shattering (orange). **i** Introgressed regions from *L. serriola* (green), *L. saligna* (yellow), and *L. virosa* (red) in cultivated lettuce. Circos[69] was used to display above results

lettuce carried the same introgressed region from *L. serriola* on LG5 (235.5–236.5 Mb).

**Genome-wide mapping of eQTL.** eQTLs may provide useful information for the regulatory network of genes and their corresponding traits. With the removal of lowly expressed genes, the expression values of 22,039 genes were retained for eQTL analysis. Their expression values were compared with 103,662 high-quality SNPs with minor allele frequencies (MAF) >5%. The EMMAX software[33] was used for association analysis of the transformed expression levels for each gene. A total of 331,706 SNPs were significantly associated with the expression of at least one gene over the Bonferroni-corrected threshold ($-\log_{10}(P) = 6.31$, $\alpha = 0.05$). Linked SNPs that are associated with the same gene were grouped into unique eQTL blocks, which are represented by the most significant SNP in the block. After grouping associated SNPs, 5311 candidate eQTL regions were identified for 4105 genes (Supplementary Data 11). Most (3299 or 80.37%) of the identified genes have only one eQTL. However, 589 genes have two eQTLs and 217 genes have three or more eQTLs.

When the positions of the eQTLs were plotted against the positions of their associated genes, a strong enrichment along the diagonal was observed (Fig. 4a). These results indicate that the majority of genes are locally regulated. Based on an analysis of the positions of the eQTL regions and their target genes, 3294 of

these eQTLs (62.02%) are considered to be local eQTLs and the remaining 2017 (37.98%) are distant eQTLs. The overall $-\log_{10}(P)$ value and explained expression variation of local eQTLs were higher than those of distant eQTLs, indicating that the local eQTLs tended to have larger effects on gene expression than distant eQTLs (Fig. 4b, c).

Distant eQTL hotspots are regions that regulate the expression of many genes. Using hot_scan[34], 49 distant eQTL hotspots were identified (Supplementary Data 12). Two regions, one on LG3 (LG3: 129896517–129898847, $P_{adjust} = 2.01 \times 10^{-230}$) and one on LG5 (LG5: 85992389–86123627, $P_{adjust} = 8.70 \times 10^{-94}$) are the most significant distant eQTL hotspots, containing 58 and 45 distant eQTLs, respectively (Fig. 4d). One or a few master regulators that regulate the expression of multiple genes (targets) are potentially responsible for these distant eQTL hotspots. Using the criteria described in "Methods" section, 267 genes from the 49 hotspots were considered as potential master regulators (Supplementary Data 13). Among these candidates, 11 have more than 20 potential targets. A gene encoding a MYB transcription factor (*MYB113*, LG5_426271) located at a distant eQTL hotspot on chromosome 5 has 47 potential targets, and the *Glutathione S-transferase* gene (*GST*, LG3_262677) located at a distant eQTL hotspot on chromosome 3 has 29 potential targets. GO analysis of the target genes of these two master regulators showed enrichment in flavonoid metabolic process (Fig. 4e).

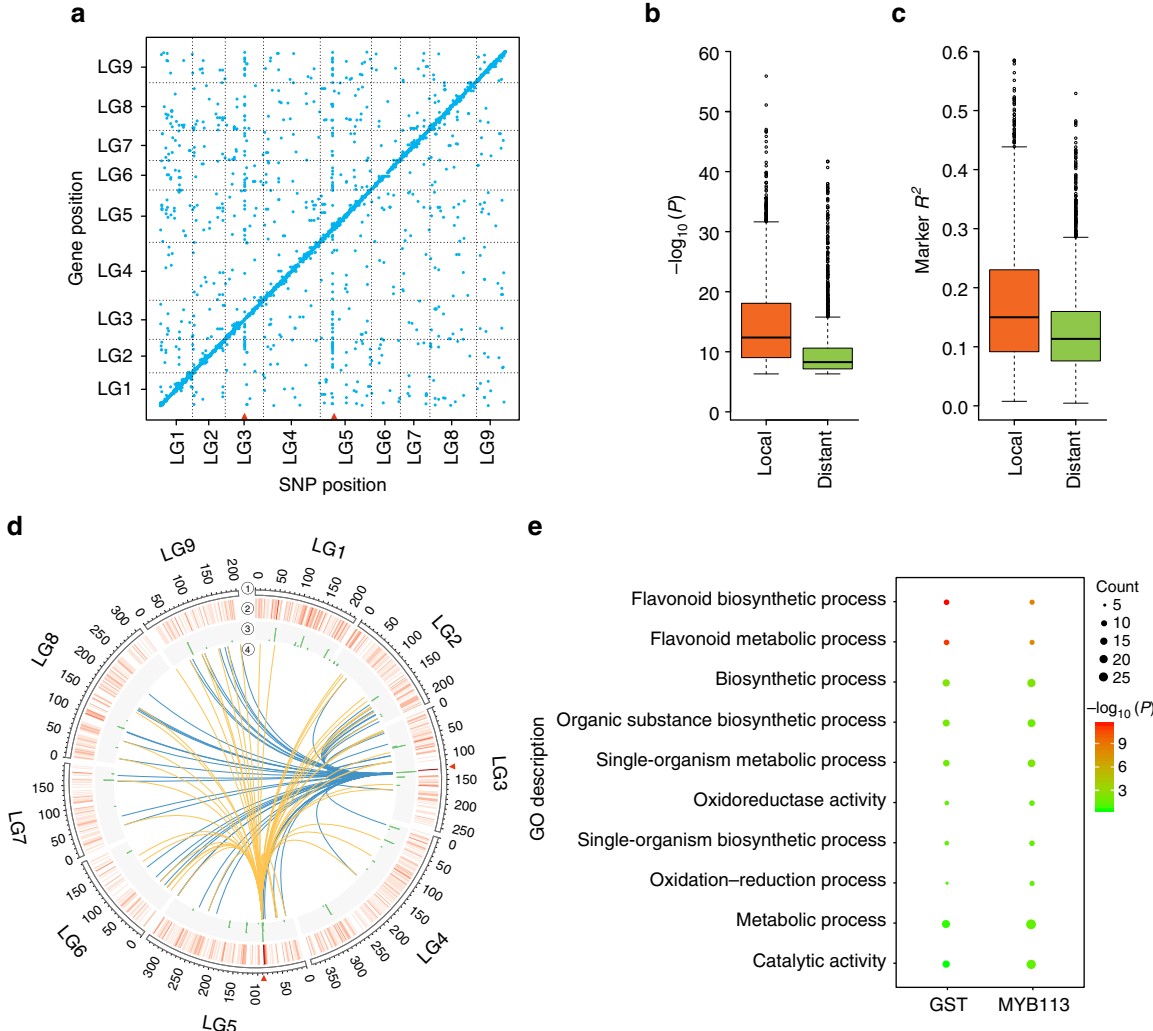

**Fig. 4** Genome-wide mapping of eQTL. **a** eQTLs identified using GWAS. The middle position of each detected eQTL is plotted against the middle position of the mapped gene. The x-axis indicates the positions of the eQTLs and the y-axis indicates the positions of the mapped genes. Each dot represents a detected eQTL. Two red triangles indicate the two most significant distant eQTL hotspots, each of which are eQTLs of dozens of genes. **b** Boxplots of the $-\log_{10}(P)$ values of local and distant eQTLs. The median value of $-\log_{10}(P)$ is 12.37 and 8.28 for local and distant eQTLs, respectively. **c** Boxplots of the effects of local and distant eQTLs. The median value of effect is 14.99% and 11.34% for local and distant eQTLs, respectively. **d** Visualization of distant eQTL hotspots using Circos[69]. Two red triangles on chromosomes 3 and 5 indicate the two most significant distant eQTLs. 1 Nine chromosomes of lettuce. 2 Heatmap that showing the counts of targets in a 2 Mb window across the genome. 3 Histogram that showing the significance (calculated by $-\log_{10}(P$ value)) of each distant hotspot. 4 Links between the two most significant distant eQTL hotspots and their target regions. **e** GO enrichment analysis of target genes of *GST* and *MYB113*, respectively

**Regulatory networks for flavonoid biosynthesis**. The flavonoid biosynthetic pathway is one of the most functionally conserved and best studied pathways in plants[5]. Three methods including eQTL mapping, gene annotation, and co-expression analysis were used to identify candidate regulators of the genes in this pathway.

A total of 153 genes potentially involved in flavonoid biosynthesis were identified by homology search and subsequently confirmed manually (Supplementary Data 14). Based on our RNA-Seq data, 148 of these genes were expressed. eQTL mapping showed that 24 of the 148 genes had at least one eQTL over the Bonferroni-corrected threshold ($-\log_{10}(P) = 6.31$, $\alpha = 0.05$). Among these 24 genes, 7 genes are regulated by only local eQTLs, 12 genes are regulated by only distant eQTLs, and 5 genes are regulated by both local and distant eQTLs. After merging the overlapping eQTLs into one region, nine candidate regions were obtained (Fig. 5a). Theoretically, this group of nine eQTLs may contribute to the variation of leaf color in lettuce. Based on our GWAS analysis, five of these eQTL regions are

coincident with regions that are responsible for the accumulation of anthocyanins in lettuce (see below).

Several eQTLs for different flavonoid-related genes were located in the same region. These data are consistent with the same regulators controlling the expression of these flavonoid-related genes. The iterative group analysis (iGA) approach[35] was used to test this possibility to identify significant regulatory groups (Supplementary Table 13) and then to construct the genetic regulatory network (Fig. 5b). Four candidate regulators were identified, which regulate the expression of 14 genes associated with flavonoid biosynthesis. These target genes include structural genes or transcription factors, such as *ANS* (Anthocyanidin synthase, LG9_787816), *CHS* (Chalcone synthase, LG2_229551), *CPC* (Caprice, LG4_332082)[36], *GL2* (Glabra 2, LG4_394790)[37], *ANL2* (Anthocyaninless 2, LG4_340440)[38], etc. Strikingly, the most significant ($P = 1.65 \times 10^{-10}$) regulator identified was *GST* (*TT19*, LG3_262677), which facilitates the transport of anthocyanin from the cytosol to the tonoplast[39].

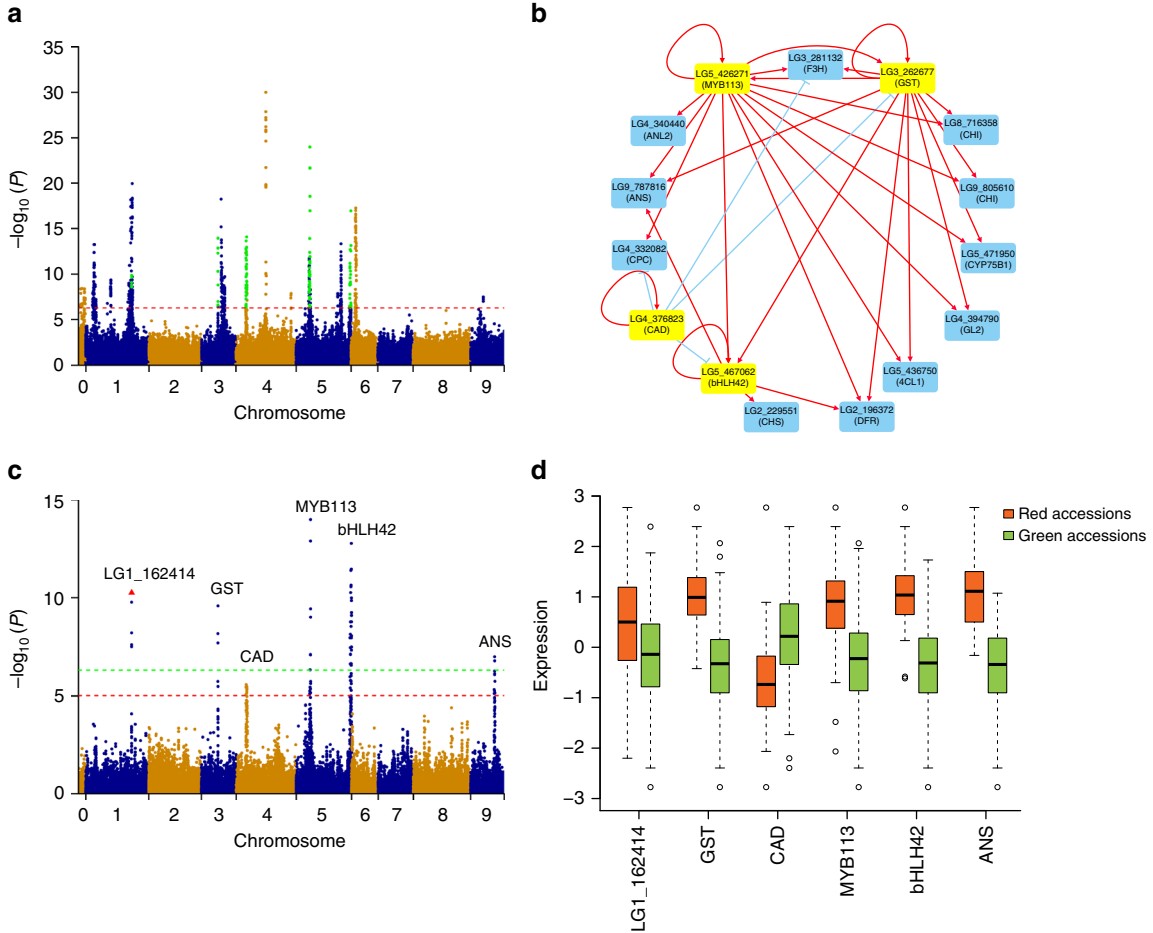

**Fig. 5** Regulatory network of genes associated with flavonoid biosynthesis. **a** Combined Manhattan plots from the eQTLs of 24 genes associated with flavonoid biosynthesis. Green dots represent the significantly associated loci that were coincident with regions identified using the GWAS for the leaf anthocyanins. Chromosome 0 represents unanchored scaffolds. The red horizontal dashed line corresponds to the Bonferroni-corrected significance threshold ($-\log_{10}(P) = 6.31$, $\alpha = 0.05$). **b** The inferred regulatory network for flavonoid biosynthesis using the iGA approach. Each node is a gene, and each edge is a predicted regulatory connection between genes. Yellow nodes represent candidate regulatory genes, and blue nodes represent the target genes. Red arrows indicate positive regulation, while blue bars indicate negative regulation. Cytoscape[70] was used to display the network. **c** Manhattan plot for leaf color. Six candidate genes are shown along the top of the significant associated signals. The red and green horizontal dashed lines correspond to the suggestive ($-\log_{10}(P) = 5.02$, $\alpha = 0.1$) and significant ($-\log_{10}(P) = 6.32$, $\alpha = 0.05$) threshold, respectively. The red triangle indicates the proposed functional site (1_125530709) in gene *LG1_162414*. **d** The identified six candidate genes were significantly ($P \leq 7.8 \times 10^{-4}$ for all six candidates) differentially expressed between red and green accessions. The expression values of each gene were normal quantile transformed

Both eQTL analysis and co-expression analysis indicated that these four candidate regulators are likely to be important regulators of flavonoid biosynthesis in lettuce.

The four candidate regulators and their 14 target genes were used as seed genes to identify other possible functionally related genes. A total of 80 genes were significantly co-expressed with at least three seed genes ($p < 0.01$). Among these genes, 74 had no reported function related to flavonoid biosynthesis (Supplementary Data 15, Supplementary Fig. 15). Of the 74 genes, 30 have eQTL support. Strikingly, the eQTLs for 29 of these 30 genes are co-located with one or more of the four candidate regulators. Based on these results, we conclude that these genes are likely associated with flavonoid biosynthesis; further study of these 29 genes will most likely advance our understanding of flavonoid biosynthesis.

**Genome-wide association of leaf color in lettuce**. The strong population structure and high LD make GWAS in lettuce difficult. However, GWAS for leaf color is predicted not to be affected considerably by population structure since red cultivars are found

in all horticultural types. FaST-LMM[40] was used to identify association signals for leaf color. Six significant loci controlling leaf color were identified at the suggestive threshold ($-\log_{10}(P) = 5.02$, $\alpha = 1$) (Fig. 5c, Table 1, Supplementary Fig. 16). Among them, five loci (locus 1–5) were co-located with eQTL for the genes in the flavonoid pathway. The GWAS results were combined with the eQTL and regulatory network analyses (see above) to identify the candidate genes responsible for leaf color variation in lettuce.

The candidate gene (*LG1_162414*) at locus 1 encodes a protein of unknown function. One local and two distant eQTLs were detected for this gene (Supplementary Fig. 17a). One of the distant eQTLs was coincident with the fifth locus. This gene also showed significantly differentially expressed between red and green leaf accessions ($P = 7.8 \times 10^{-4}$, Fig. 5d). One SNP (1_125530709) in the 6th exon was found to cause a premature stop codon, and it is likely responsible for the functional variation of this gene. The candidate gene (*LG3_262677*) at locus 2 encodes GST. Three eQTLs were detected for the *GST* gene, including one local and two distant eQTLs (Supplementary Fig. 17b). The two

**Table 1 Summary of SNPs associated with the leaf color at the suggestive threshold**

| Loci ID | Chromosome | Region^a | Coincident with eQTL | lead SNP | −log₁₀(P) | PVE (%) |
|---|---|---|---|---|---|---|
| 1 | LG1 | 125530637–125531891 | YES | 125530709 | 10.25 | 19.02 |
| 2 | LG3 | 129896518–129898737 | YES | 129898324, 129898534, 129898572 | 9.59 | 18.62 |
| 3 | LG4 | 49271455–53002347 | YES | 49497812 | 5.58 | 7.98 |
| 4 | LG5 | 83217466–86153183 | YES | 86123627, 86123633, 86123651 | 13.99 | 28.50 |
| 5 | LG5 | 331079106–337623748 | YES | 336591721 | 12.77 | 27.52 |
| 6 | LG9 | 151490793–153084023 | NO | 152892248 | 7.01 | 14.17 |

PVE, explained phenotypic variation
^aThe associated regions were identified based on the highly linked significant SNPs at the suggestive threshold ($-\log_{10}(P) = 5.02$, $\alpha = 1$)

distant eQTLs were linked with the third and fourth locus, respectively (see below). The expression of this gene in red leaf accessions was significantly higher than that in green leaf accessions ($P < 1.3 \times 10^{-9}$, Fig. 5d).

Using a similar approach, the candidate genes from the other loci were identified. All of the candidate genes exhibited significantly differential expression levels between red leaf and green leaf accessions ($P < 0.01$, Fig. 5d). For the third locus, the candidate gene is *CAD* (LG4_376823), encoding a cinnamyl-alcohol dehydrogenase that contributes to secondary metabolism[41]. Only one local eQTL was identified for this gene (Supplementary Fig. 17c). The higher expression of this gene in green leaf accessions than that in red leaf accessions ($P = 2.2 \times 10^{-6}$, Fig. 5d), indicates it may have a negative effect on leaf color in lettuce. For the fourth locus, the strongest candidate gene is a *MYB* (*MYB113*, LG5_426271). As described above, *MYB113* is one of the master regulators that we identified. One local eQTL and one distant eQTL were detected for this gene (Supplementary Fig. 17d). The distant eQTL was linked to the second locus. For the fifth locus, we concluded that the strongest candidate is a gene encoding a bHLH transcription factor (*bHLH42*, LG5_467062). One local eQTL and four distant eQTLs were detected for *bHLH42* (Supplementary Fig. 17e). The four distant eQTLs were coincident with the locations of the first to fourth loci, respectively. For the sixth locus, only one gene *ANS* (*TT18*, LG9_787816) in the interval is associated with flavonoid biosynthesis. Three distant eQTLs of the *ANS* gene overlapped with the second, fourth, and fifth locus, respectively (Supplementary Fig. 17f).

The loci potentially contributing to color variation in lettuce were then analyzed genetically. Segregating populations that segregated for individual locus were successfully obtained for loci 3, 4, 5, and 6. Genetic markers were designed for these regions and used to screen these four populations. The four loci co-segregated with leaf color in each of these four populations, confirming the GWAS results (Supplementary Note 6, Supplementary Table 14).

## Discussion

In this study, we performed RNA sequencing of 240 *Lactuca* spp. genotypes, generating a total of 1.1 million high-quality SNPs and 24,042 gene expression profiles. Our results suggested that lettuce undergo a single domestication from *L. serriola*, and it was estimated to be domesticated 10,829 y B.P. (95% CI = 10,391–13,005 y B.P.) in the Middle East and the Fertile Crescent. Interestingly, a *L. serriola* accession (CGN04799, with code W44 in this study) is more similar to cultivated lettuce than the other wild accessions studied. According to the Centre for Genetic Resources (CGN), this accession was collected 20 km south of Al-Fallujah in Iraq, this is within the Fertile Crescent where lettuce was domesticated. The W44 is not different from other wild accessions in leaf shape, flowering time, shattering, although it has a lower density of

spines on stems. The close relationship between the wild accession W44 and cultivated lettuce may reflect its location near the center of domestication. This hypothesis could be tested in the future by analysis of additional genotypes collected from Al-Fallujah and its surrounding regions.

To successfully identify loci controlling traits in lettuce populations, the trait of interest should exhibit a broad variation within sub-populations. For example, the lettuce population in this study exhibited a broad variation in leaf color in all horticultural types. In this study, a model that considers population structure was used to successfully identify six loci associated with leaf color in lettuce, and four of them have been genetically verified, showing success of GWAS in lettuce. The high LD makes it challenging to identify a candidate gene in a detected locus. Our results showed that the application of eQTL and gene expression network analysis greatly improved the chances to identify a candidate gene in GWAS.

eQTL analysis in this study identified nine loci in lettuce contributing to expression variation in genes associated with anthocyanin biosynthesis. However, only five of them were shown to contribute leaf color variation. In the future, it will be interesting to investigate the roles of the other four eQTLs in the quantity and diversity of flavonoids in lettuce. Expression network analysis may also identify novel genes in the anthocyanin biosynthesis. Our analysis yielded dozens of genes that are potentially associated with flavonoid synthesis, some of them have unknown functions or functions not related to flavonoid biosynthesis. Future experiments will test whether these genes contribute to the accumulation of flavonoids and may provide insight in flavonoid biosynthesis and its regulation in lettuce.

## Methods

**Plant materials and RNA sequencing.** Most of *Lactuca* materials were ordered from the USDA GRIN website (http://www.ars-grin.gov/) and the Centre for Genetic Resources, the Netherlands (CGN) website (http://www.wageningenur.nl/); some were bought from commercial websites or Chinese seed market and some accessions of *L. serriola* were collected in natural populations in China by H. Kuang. All *Lactuca* materials (more than 1000 accessions) were sown in November, 2014 in greenhouses on the campus of Huazhong Agricultural University, Wuhan, China. Based on the phenotypic variation, 240 accessions were chosen for genetic analysis in this study.

Total RNA was extracted from the top fully expanded leaves from 2.5-month-old plants using the TRIzol reagent (Invitrogen) according to the manufacturer's protocol. The non-directional paired-end RNA-Seq library of each accession was prepared following the Illumina TruSeq RNA sample preparation kit, version 2. The libraries were sequenced on the Illumina HiSeq 2500 platform to obtain 125-bp paired-end reads.

**Mapping and SNP calling.** Raw RNA-Seq data were filtered to remove sequencing adapters and low-quality bases using Trimmomatic (version 0.33)[42]. The filtered fastq files were aligned to lettuce genome (version 8)[17] using the STAR software (version 2.4.2a) in the 2-pass mapping mode[43]. After alignment, Picard tools (version 1.139, http://broadinstitute.github.io/picard/) were used to remove PCR duplicates according to the mapping coordinates. Finally, a single BAM file was generated for each accession and was used for further analysis.

To detect SNPs in the population, raw SNPs were called using the mpileup command in the SAMtools package (version 1.2)[44]. BCFtools was further used to

filter potentially false SNPs using the following criteria: (1) mapping quality, total sequencing depth, and SNP quality each had to be equal to or greater than 30; (2) each SNP had to be more than three base pairs away from an InDel; (3) each SNP had to be biallelic; (4) for homozygous genotypes, supporting reads had to be greater than 3 and the SNP quality had to be greater than 20; (5) for heterozygous genotypes, supporting reads for the reference and alternative allele had to be equal to or greater than 2 and the genotype quality had to be greater than 20; and (6) for sites that failed to pass, based on the above criteria, the individual genotypes of these sites were assigned as missing.

**Functional annotations and enrichment analyses.** The effects of SNP on genes was predicted using SnpEff (version 4.1l)[45]. The results were parsed using in-house Perl scripts. Gene Ontology (GO) annotations were assigned using the Trinotate pipeline[46]. Protein domain families were assigned by Pfam database[47] with an e-value cutoff as 1e−5. GO term and protein domain enrichment analysis were performed using agriGO (v2.0)[48] and FuncAssociate (v3.0)[49] at $P < 0.05$.

**Imputation of missing data.** The fillGenotype software[18] that is based on the k-nearest neighbor algorithm was used to impute missing genotypes. In order to get the optimal imputation accuracy and filling rate, the following steps were used: (1) SNP sites were divided into ten categories according to the missing rate, which ranged from 10 to 90%; (2) 1% SNP sites for each category were randomly masked as missing genotypes to measure the imputation accuracy and filling rate; (3) the imputation was done using the fillGenotype software using the following parameters: $w$ (20, 30, 50, 65, 80), $p$ (−5, −7, −9, −11), $k$ (3, 5, 7, 9), and $r$ (0.5, 0.6, 0.7, 0.8). These parameters yielded 320 sets of parameters for each category; (4) after testing 320 combinations of parameters, the best imputation accuracy and filling rate were obtained for each category under certain parameters; (5) the best results of each category were plotted and compared to each other. Certain missing rates with optimal imputation accuracies and filling rates were selected; and (6) SNP sites with decided missing rates and parameters were used to impute missing genotypes.

**Population genetic analysis.** The RAxML software[50] was used to construct maximum-likelihood trees using SNPs without imputation, with the following parameters: -m ASC_GTRGAMMA --asc-corr = lewis -f a -p 23 -x 123 -# 100. A nonparametric bootstrap analysis was performed, with 100 bootstrap replicates. Accession "W17" which belongs to *L. virosa* was set as an outgroup. The final tree was visualized using iTOL software[51].

The EIGENSOFT software package[52] was used to perform PCA on individual genotypes. SNPs with missing rates < 20% were used to perform PCA analysis. The first two components were plotted for the lettuce accessions.

The STRUCTURE program was used to infer population structure[19]. The "admixture model" implemented in STRUCTURE was used to estimate the ancestry and admixture proportions in each individual. SNPs with missing rates <10% and MAF >5% were used. For each K value that ranged from 1 to 20, STRUCTURE was run 20 times with an admixture model and 10,000 burn-in and MCMC replicates. The final results were imported into CLUMPAK[53] to estimate the most likely number of sub-populations and to graphically represent the sub-population membership of each accession.

LD ($r^2$) was calculated for all pairs of SNPs after imputation within 5000 kb using PopLDdecay software (version 1.01, https://github.com/BGI-shenzhen/PopLDdecay). The following parameters were used: -MaxDist 5000 -MAF 0.05 -Het 0.88 -Miss 0.25. The maximum value of $r^2$ was calculated based on SNP pairs within 1 kb. For each group, a LD decay curve was plotted based on $r^2$ and the distance between pairs of SNPs.

**Demographic modeling.** To reconstruct the domestication history of lettuce, we used the joint site frequency spectrum (SFS) approach implemented in *fastsimcoal2* (version 2.5.2.21)[20]. To minimize the bias in demographic inferences due to selection, only neutral sites (4DTV, four-fold synonymous transversion) were used for this analysis. Furthermore, SNPs that had missing data before imputation in populations were excluded, resulting in 46,002 segregating sites of the 2,073,047 total 4DTV sites without missing data in the final data. The folded SFS was created using a modified script from $\delta a \delta i$[54]. First, one-population models were run for all groups including the wild ancestor *L. serriola* and four individual horticultural types (butterhead, crisphead, romaine, and stem lettuce). Second, each horticultural type was analyzed together with *L. serriola* in two-population models. Third, three leafy horticultural types (butterhead, crisphead, and romaine) were analyzed jointly with *L. serriola* in four-population models. Finally, stem lettuce was incorporated into five-population models. In four- and five-population models, sample sizes were projected down to 18, 14, 14, 14, and 14 for *L. serriola*, butterhead, crisphead, romaine, and stem, respectively, due to the one million entry limitation of the SFS files that *fastsimcoal2* can process. For each model, at least 25 independent runs with varying starting points were conducted to determine the parameter estimates leading to the maximum likelihood. Each run consisted of 20–40 rounds (−l 20, −L 40) of parameter estimation using the expectation-conditional maximization algorithm with a length of 100,000 simulations per likelihood estimation (−n 100,000, −N 100,000). The best model was determined based on the maximum

value of the likelihoods and the akaike information criterion[55]. To obtain parameter confidence intervals (CIs), 100 bootstrap data sets were created by sampling with replacements from the 4DTV sites. Parameters were estimated from 50 independent optimizations for each bootstrapped data set. To calculate absolute values of population size and divergence time, we assumed a mutation rate of $4 \times 10^{-8}$ per bp per generation.

**Transcriptome analysis.** The expected read counts and fragments per kilobase per million reads (FPKM) for each gene were calculated using StringTie[56], based on the alignment to the reference genome.

To calculate the expression diversity in a population, the CV was calculated for each gene as a standard deviation (SD) of FPKM divided by mean FPKM in the population using an in-house Perl script.

**Selection and introgression analysis.** The combination of cross-population composite likelihood ratio (XP-CLR)[57] and $\pi$ ratio methods was used to identify regions associated with domestication in lettuce. We first performed a genome scan using XP-CLR between *L. serriola* and cultivated lettuce. Genetic distances between adjacent SNPs were calculated using a previous genetic map by assuming a uniform recombination between mapped markers[58]. The program XP-CLR was run for each chromosome with parameters "-w1 0.005 100 2000 1 -p1 0.7". Mean likelihood score was calculated using 20 kb sliding windows with a step size of 10 kb across the genome. Adjacent windows with XP-CLR values in the top 20% were grouped into a single region. Merged regions across the genome with XP-CLR values in the top 5% were identified. We further calculated $\pi$ ratio between *L. serriola* and cultivated lettuce using 20 kb sliding windows with a step size of 10 kb. Windows with the top 50% of highest $\pi$ ratios were selected and merged into regions. Finally, regions identified by both XP-CLR and $\pi$ ratio methods were considered to be associated with domestication.

The combination of population-based integrated haplotype score (PiHS) and $\pi$ ratio methods[28] was used to identify regions under selection in different horticultural types. To detect the selective signals for a horticultural type, cultivated accessions belonging to that horticultural type were grouped as the object population and the remaining cultivated accessions were grouped as the reference population. We first calculated PiHS for each SNP across the genome between object and reference populations. The normalized (z-score) value of the PiHS was then calculated using 200-kb sliding windows with a step size of 100 kb. Regions with z-score larger than 2.33 ($P \leq 0.01$) were merged. We further calculated $\pi$ ratio between object and reference populations using 200 kb sliding windows with a step size of 100 kb. Windows with the top 50% of highest $\pi$ ratios were selected and merged into regions. Finally, regions identified by both PiHS and $\pi$ ratio methods were considered to be under selection.

Identification of introgressed regions from wild *Lactuca* species (*L. virosa*, *L. saligna*, and *L. serriola*) to cultivated lettuce was performed using a combined approach of likelihood ratio test[10] and phylogenetic analysis. Briefly, the cultivated group was compared with three wild *Lactuca* groups. SNPs with missing data and heterozygous genotypes were excluded from further analysis. For each cultivated lettuce, the ratio of shared genotype in cultivated group versus each wild *Lactuca* group was calculated in 200 kb sliding windows with a step size of 20 kb. Regions with ratio ≤0.5 and SNP number ≥20 were merged and further used for phylogenetic analysis. FastTree2[59] was used to build the maximum-likelihood phylogenetic bio-neighbor joining tree for each region, and the R package ape[60] was used to display the tree file. Then each phylogenetic tree was manually checked whether the putative introgressed cultivars were located within the putative corresponding donor wild *Lactuca* group. Finally, regions with distances <200 kb were merged.

**eQTL analysis.** Only cultivated accessions (including RIL lines) were used for eQTL mapping. Genes with a median FPKM value equal to 0 were excluded from the eQTL mapping. To identify outlier samples, PCA was performed to investigate the first three principal components (PCs), which together explain 92.79% of the variance in the expression data. We removed seven samples (L3, R1, R31, R40, R46, S15, and S32) that were >2.5 SD from the mean in any of the first three PCs, yielding a total of 180 samples ($n = 180$) for further analysis. To obtain a normal distribution of expression values for each gene, normal quantile transformation of expression values for each gene was performed using the qqnorm function in R[61,62]. Genome-wide associations of transformed expression were estimated using EMMAX[33]. The BN matrix of all SNPs calculated using the EMMAX was used to control population structure and kinship among individuals. The hidden and confounding factors that contributed to the variability of expression were identified using the PEER program[63]. Ten factors were treated as additional covariates to increase the detection power for eQTL. The Bonferroni test criterion at $\alpha = 0.05$ was used as a threshold for significance of associations between SNPs and traits. In this study, the Bonferroni-corrected threshold for the $P$ value was $0.05/(\text{total SNPs}) = 3.66 \times 10^{-6}$, with a corresponding $-\log_{10}(P)$ value of 6.31. An R script was used to generate quantile–quantile (q–q) plots and Manhattan plots from $P$ values of each SNP.

The 95% quantile of $r^2$ distribution for randomly selected inter-chromosomal SNP pairs was considered as the background level of LD[64]. The background level of

LD and physical distance at which background LD has decayed were used to assign adjacent significant SNPs into a unique eQTL region, which was 0.1 and 4.7 Mb in this study, respectively. The associated SNPs were grouped into one region if the distance between the two neighboring SNPs was <4.7 Mb and $r^2$ >0.1. The grouped region with at least three significant SNPs was considered as an eQTL block. Otherwise these SNPs were considered as false positive signals. SNPs with the minimum $P$ value in an eQTL block were considered as the lead SNPs. The lead SNP in each block was chosen as a marker for the locus for further analysis and is not necessarily a causal variant.

To identify the genetic mechanisms underpinning the eQTLs, the positions of the eQTLs and their associated genes were compared. An eQTL was defined as a local eQTL if the region of the eQTL spans the associated gene. An eQTL was defined as a distant eQTL if the eQTL is located in a distinct region relative to its target gene.

To identify distant-eQTL hotspots, the hot_scan software[34] was used. The window size and the Benjamini and Yekutieli adjusted $P$ values were set to 400 kb and 0.05, respectively.

To identify the candidate master regulators, the following criteria were used: (1) the candidate regulator gene should be located in a distant eQTL region; (2) the candidate regulator should have a local eQTL; and (3) a correlation should exist between the target gene and the master regulator. The pair-wise Pearson's correlation coefficient (PCC) for the target gene and the candidate master regulator was calculated using $\log_{10}$ (FPKM+1) values. To identify significant correlations, the threshold PCC value was set at either 0.39 or −0.28 ($P$ < 0.05) based on the permutations.

**Network construction.** The genetic regulatory network was constructed using genes known to be involved in a pathway and their potential regulators, following previous studies with modifications[65,66]. Briefly, overlapping distant eQTLs for different genes of a pathway were merged. Each gene from the merged eQTL region was used to calculate the PCC with the relevant target genes (i.e., genes in the pathway of interest). Based on the sorted PCC values, an iGA algorithm[35] was used to define the probability of change (PC) value for each gene in the merged eQTL region. If a gene in the merged eQTL region had a PC value of <0.01/(total gene number in the region), we classified this gene as a candidate regulator that may regulate those genes contributing to the significant PC value.

To detect additional genes associated with a pathway, the regulators and target genes identified using the iGA approach were used as seed genes. Pair-wise PCC values were calculated between the seed genes and all of the other genes in the genome. Two genes were considered to be related to each other if their absolute PCC value was larger than 0.44 ($P$ < 0.01, based on 1000 permutations). If a gene was related with at least three seed genes, it was considered to be associated with the pathway of interest.

**Phenotyping and association analysis.** Two replicates of cultivated accessions ($n$ = 187) were planted in an experimental field using a randomized complete block design in Wuhan, Hubei Province, China in 2015. Leaf color was classified as either green or red. Association analysis was conducted using the FaST-LMM[40] software. Suggestive ($-\log_{10}(P) = 5.02$, $\alpha = 1$) and significant ($-\log_{10}(P) = 6.32$, $\alpha = 0.05$) thresholds were used to identify significantly associated SNPs[67,68]. Highly linked significant SNPs may be considered as one single signal, as described above for eQTLs detection.

**Data availability.** The sequencing data for this project have been deposited at the NCBI Sequence Read Archive (SRA) under project PRJNA394784, accession SRP113265. The authors declare that all other data supporting the findings of this study are included in the main manuscript file or Supplementary Information or are available from the corresponding author upon request.

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

## Acknowledgements

Most of the materials in this study were kindly provided by the Germplasm Resources Information Network, USDA and Centre for Genetic Resources, The Netherlands. This work was supported by Chinese Natural Science Foundation awards # 31471162 and 31572131. Work in the RWM lab was supported by a NSF Plant Genome Program award # DBI0820451 and the National Research Initiative (NRI) Specialty Crops Research Initiative (SCRI) of the USDA Cooperative State Research, Education and Extension Service (CSREES) awards # 2010-51181-21631 and 2015-51181-24283.

## Author contributions

H.K. designed the experiment. L.Z. performed the data analysis. W.S. and R.T. did genetic analysis. W.Z., J.C., P.W., L.Z., R.T., C.Y., and Y.J. planted the population, scored plant phenotype, and extracted RNAs. D.L., M.-J.T., S.R.C.-W. and R.W.M. prepared the lettuce genome sequences. L.Z. wrote the manuscript, with help from H.K., R.W.M., R.M. L. and D.L.

## Additional information

**Competing interests:** The authors declare no competing financial interests.

