## [Peer Review File · Nature Communications]

Reviewers' comments:

Reviewer #1 (Remarks to the Author):

Lettuce (*Lactuca sativa* L.) is an important vegetable worldwide. This manuscript studied the genome evolution of lettuce and the regulation of flavonoid biosynthesis using a large-scale RNA sequencing. The authors discovered a single domestication event and identified selective sweeps using RNA-seq data. They identified some candidate loci or genes associated with flavonoid biosynthesis (anthocyanins). The discoveries and dataset are very valuable for lettuce biology and breeding. Nevertheless, I have a few comments for the manuscript.

1. The study of flavonoid biosynthesis is a highlight of the study. I hope the GWAS and transcriptome analyses can be supported by linkage analyses. The genotyping and phenotyping of few F2 population are not difficult.
2. The comparison of the importance of local versus distant eQTLs is invalid (Page 12). As the sequencing focused on the coding region, the distant eQTLs at promoter regions and intergenic regions have therefore not been surveyed in the study.
3. Figure 3 and Figure 4 are from one topic and the content can be merged. AIn Introduction part, the classification of horticultural types is based on their morphological characteristics. Is Leafy, stem, oil vegetable based on these types? Need be described.
4. In the regions under artificial selection, what is genome-wide nucleotide diversity (π) for each group/type? The authors could merge two methods, XP-CLR and diversity reduction, to identify selective sweeps.
5. In Distant eQTL hotspots and master regulators from Results, "Most of their target genes are associated with flavonoid metabolism". This has a vague idea. Authors should provide the significance value or the number of genes associated with flavonoid metabolism, such as GO enrichment or ratio.
6. Different r^2 value (0.3 or 0.1) were used for LD threshold, and the same question for α (1 or 0.05). The author should make these more clear and reasonable.

Reviewer #2 (Remarks to the Author):

The manuscript by Zhang and colleagues analyses RNAseq data from a large collection of lettuce accessions to explore the history of domestication and diversification of this species.

The authors first generate RNAseq data for a collection of 240 accessions of cultivated lettuce and of its wild relatives. SNP data from this dataset allow them to distinguish most horticultural types of lettuce, and suggest that they all derive from a single domestication event from *Lactuca serriola*.

They then use transcript abundance data to identify changes in gene regulation between wild and cultivated lettuce, and to identify eQTL.

Finally, they perform a GWA for leaf colour, and combine it with eQTL and expression data to generate a comprehensive analysis of the flavonoid pathway controlling leaf pigmentation in cultivated lettuce.

The manuscript is written clearly and presents interesting insights into the domestication history of lettuce. I was also particularly impressed by how the authors successfully combined different approaches (GWA, eQTL) to identify genes involved in leaf colour variation and their regulatory networks.

While the manuscript includes many detailed analyses, their implications are often only briefly

mentioned in the Results section, and are not mentioned at all in the Discussion. I think the manuscript would greatly benefit from a more developed and organic discussion of the experimental results. As an alternative, some of the analyses that are less relevant to the main findings of the manuscript could be moved to the supplemental material. For example:

- The authors point out that LHY is identified as relevant in both the AEC and domestication analyses. Do they have a hypothesis on what is the possible effect of altering LHY regulation during domestication? Similarly, the authors mention the fact that LD and NPR3 have deleterious SNPs in lettuce accessions (how many), but do not elaborate on why they highlight these two genes. An analysis of which gene families or functional categories are affected might be more informative (I would expect, as found in other species, that R genes and F box genes would be over-represented).
- About half of the genes with the strongest AEC patterns (including LHY) have no or almost no genes co-expressed in cultivated lettuce. That is the case for only one gene in wild lettuce. Is that expected? What is the reason for this pattern? Also, the conclusions the authors draw from the results presented in this section (line 257-258: "transcriptional networks became rewired in cultivated lettuce") seem overly vague, and should either be expanded or removed.
- Did the authors find any evidence of introgression during domestication from other wild *Lactuca* species?
- The results shown in Figure 2 should be further discussed; in particular, there is no comment in the text on the dispersion history of other horticultural types besides stem lettuce, or on the meaning of the migration rates (MIGS and MIGC) reported in Figure 2a.

Additionally, in the Discussion section, the authors twice use unpublished data to support their results (line 433 and 456). These data would be a great addition to the manuscript, as they would strengthen the authors' conclusions. Otherwise, these mentions should be avoided.

The labels in several of the figures are too small and very hard to read even when magnified. Even in the original Word file, the resolution of Supplementary Figure 6 and 10 is too low for the image to be understood. Contrast should be increased for the PCA plots in figure 1b, since in several cases the individual data points are difficult to see.

Other points:

- Introduction: The focus on stem lettuce in China is a bit confusing at first, since the fact that stem lettuce is very different from other types and is predominantly cultivated in Asia is only mentioned in the discussion. That information should probably be moved to the introduction
- line 90: it would be helpful to move here some of the information on what the 240 lines are and how they have been selected; most people wouldn't read the Material and Methods first and might be therefore confused with respect to what is what. Also, the main text never mentions the RILs and how they have been considered in the analyses.
- Line 132-134: I am not sure what are the implications of this statement. While domesticated accessions are more abundant, they clearly have much less genetic variability than wild accessions.
- Line 254-256: Supplementary Figure 13: is the difference in average number of genes co-expressed with AEC genes in cultivated lettuce and *L. serriola* significant? There seem to be a lot more variation for that in domesticated lettuce (as mentioned before, almost half of the AEC genes have no or almost

no gene co-expressed with them, and that is compensated by a smaller number of AEC genes with >1000 co-regulated genes). More in general, variation in the average number of co-regulated genes does not seem the most meaningful parameter to compare genetic networks.

- Lines 367-427: I found this section really interesting, but all the connections between different loci are a bit hard to follow at times. If at all possible, it would be useful to condense all these information in a figure or table (they are currently scattered over several figures in the main text and supplementary material). I also found the expression data for the five candidate loci to be interesting results, and they could probably be moved to the main text

- Line 373: I guess that is not meant to be $\alpha = 1$ (the same value is reported also in line 684)

- Line 373: according to Table 3, it should read "(Locus number 1 to 5)

- Line 468: These examples are not really meaningful and should be removed; bHLH and WD40 are widespread protein motifs, and MYB transcription factors regulate many other processes besides flavonoid biosynthesis.

- Line 863: it should read "Supplementary Table 9".

- In Figure 1a, one individual of *L. serriola* appears to be much more similar than the others to intermediate and cultivated lettuce. Is that simply a mis-identified hybrid plant, or could that be (close to) the original population from where lettuce was domesticated?

- Most of the main information in the Manhattan plot in Figure 4 is already presented in Figure 3a. It would make sense to fully combine the Manhattan plots in figures 3 and 4, to allow direct comparison (and move panel 4b to the Supplementary information).

Responses to reviewers' comments:

Reviewer #1:

Lettuce (*Lactuca sativa* L.) is an important vegetable worldwide. This manuscript studied the genome evolution of lettuce and the regulation of flavonoid biosynthesis using a large-scale RNA sequencing. The authors discovered a single domestication event and identified selective sweeps using RNA-seq data. They identified some candidate loci or genes associated with flavonoid biosynthesis (anthocyanins). The discoveries and dataset are very valuable for lettuce biology and breeding. Nevertheless, I have a few comments for the manuscript.

1. The study of flavonoid biosynthesis is a highlight of the study. I hope the GWAS and transcriptome analyses can be supported by linkage analyses. The genotyping and phenotyping of few F2 population are not difficult.

RESPONSE:

We have included the results of linkage analyses in the revised manuscript. Four loci have been genetically verified in segregating populations.

We agree that the linkage analysis results make the conclusions more convincing. There are several loci controlling the color variation in lettuce. Sometimes it is challenging to construct a population that only one of the color controlling genes is segregating. Using different crosses and sub-populations, we succeeded in analyzing four loci; these linkage analyses have been added to the revised manuscript.

2. The comparison of the importance of local versus distant eQTLs is invalid (Page 12). As the sequencing focused on the coding region, the distant eQTLs at promoter regions and intergenic regions have therefore not been surveyed in the study.

RESPONSE:

The reviewer is correct in that the promoter regions and intergenic regions were not surveyed in our study. When the importance (*i.e.* proportion of explained expression variation) of an eQTL was calculated, the lead SNP in the detected region was used to represent the locus (the candidate gene). The lead SNP was only a marker and not necessarily the causal variant. For a local eQTL, the causal mutation is likely located in the promoter of the gene, whose sequence was not available in this study. For a distant eQTL (such as a transcription factor), the causal mutation can be either located in the promoter region (controlling its expression) or in the coding region (controlling protein function). Since the lead SNP was used for both local and distant eQTLs, we think such comparison is valid and the results were kept in the revised manuscript.

To clarify, the following sentence was added in the Methods, section of “eQTL analysis”: “The lead SNP in each block was chosen as a marker for the locus for further analysis and is not necessarily a causal variant”.

3. Figure 3 and Figure 4 are from one topic and the content can be merged. In Introduction part, the classification of horticultural types is based on their morphological characteristics. Is Leafy, stem, oil vegetable based on these types? Need be described.

RESPONSE:

We have combined Figure 3 and Figure 4 into one Figure (Figure 5a-c in the revised manuscript). As suggested by another reviewer, the expression data of the six candidate genes between red and green accessions are moved from the supplementary file to the Figure 5d.

“Leafy”, “stem” and “oilseed” lettuce were classified based on organs that are harvested and consumed. “Leafy” lettuce is a collective name of all lettuce types whose leaves are harvested and consumed. Leafy lettuce includes horticultural types of looseleaf, romaine, butterhead, crisphead, which are morphologically diverse. The classification of horticultural types is added to the introduction of the revised manuscript.

4. In the regions under artificial selection, what is genome-wide nucleotide diversity (π) for each group/type? The authors could merge two methods, XP-CLR and diversity reduction, to identify selective sweeps.

RESPONSE:

Thanks for the suggestions. In the revised manuscript, we have made appropriate changes. The combination of XP-CLR and π ratio methods was used to detect selected regions between *L. serriola* and cultivated lettuce, *i.e.* domestication. The combination of PiHS and π ratio methods was used to identify regions under selection in one horticultural type compared with other horticultural types.

5. In Distant eQTL hotspots and master regulators from Results, “Most of their target genes are associated with flavonoid metabolism”. This has a vague idea. Authors should provide the significance value or the number of genes associated with flavonoid metabolism, such as GO enrichment or ratio.

RESPONSE:

We have added the GO enrichment analysis of the target genes in the revised manuscript. The results showed that the two master regulators’ target genes have enrichment in flavonoid metabolic process.

6. Different r^2 value (0.3 or 0.1) were used for LD threshold, and the same question for α (1 or 0.05). The author should make these more clear and reasonable.

RESPONSE:

For r^2 value of 0.3:

Various methods have been used to estimate the threshold of LD, and there is no consistent approach in the literature¹. For studies that use large-scale SNPs identified by next-generation sequencing (NGS) technology, a commonly used method to define the threshold of LD decay is half of the maximum r^2 value²⁻⁶. To facilitate comparison among different species, we used this method to estimate LD decay of different

lettuce groups/types. We have made appropriate changes in the revised the manuscript.

For r^2 value of 0.1:

We used background level of LD and physical distance at which background LD has decayed to assign adjacent significant SNPs into a unique eQTL region. The background LD is a threshold that determine whether LD between two SNPs is due to genetic linkage⁷. The 95% quantile of r^2 distribution for randomly selected inter-chromosomal SNP pairs was considered as the background level of LD. In our study, the 95% quantile of r^2 distribution was 0.105049 and 0.1 was set as the background level of LD. And the physical distance at which background LD has decayed was 4.7 Mb in the eQTL mapping population. We have added the above information to the methods section of “eQTL analysis”.

For different α value (1 or 0.05)

These two Bonferroni-corrected thresholds have been widely used to control genome-wide type I error rate in GWA studies. For example, Chen *et al.*, Si *et al.* and Gao *et al.* used $\alpha = 0.05$ as the threshold⁸⁻¹⁰; Li *et al.*, Liu *et al.*, Wen *et al.*, and Yang *et al.* used $\alpha = 1$ as the threshold¹¹⁻¹⁴. Several studies^{15,16} used these two as the suggestive ($\alpha = 1$) and significant ($\alpha = 0.05$) thresholds.

In our study, we used the significant threshold for eQTL mapping ($-\log_{10}(P) = 6.31$, $\alpha = 0.05$), and used the suggestive threshold for GWA of leaf color ($-\log_{10}(P) = 5.02$, $\alpha = 1$). For GWA of leaf color, a signal was detected for locus 3 on chromosome 4 when using $\alpha = 1$, which was subsequently confirmed genetically. However, using $\alpha = 0.05$ failed to identify this locus, generating a false negative result.

To be consistent, both $\alpha = 0.05$ (significant threshold) and $\alpha = 1$ (suggestive threshold) in Bonferroni-correction was used in the GWAS of leaf color in the revised manuscript.

Reviewer #2:

The manuscript by Zhang and colleagues analyses RNAseq data from a large collection of lettuce accessions to explore the history of domestication and diversification of this species.

The authors first generate RNAseq data for a collection of 240 accessions of cultivated lettuce and of its wild relatives. SNP data from this dataset allow them to distinguish most horticultural types of lettuce, and suggest that they all derive from a single domestication event from *Lactuca serriola*.

They then use transcript abundance data to identify changes in gene regulation between wild and cultivated lettuce, and to identify eQTL.

Finally, they perform a GWA for leaf colour, and combine it with eQTL and

expression data to generate a comprehensive analysis of the flavonoid pathway controlling leaf pigmentation in cultivated lettuce.

The manuscript is written clearly and presents interesting insights into the domestication history of lettuce. I was also particularly impressed by how the authors successfully combined different approaches (GWA, eQTL) to identify genes involved in leaf colour variation and their regulatory networks.

While the manuscript includes many detailed analyses, their implications are often only briefly mentioned in the Results section, and are not mentioned at all in the Discussion. I think the manuscript would greatly benefit from a more developed and organic discussion of the experimental results. As an alternative, some of the analyses that are less relevant to the main findings of the manuscript could be moved to the supplemental material. For example:

- The authors point out that LHY is identified as relevant in both the AEC and domestication analyses. Do they have a hypothesis on what is the possible effect of altering LHY regulation during domestication? Similarly, the authors mention the fact that LD and NPR3 have deleterious SNPs in lettuce accessions (how many), but do not elaborate on why they highlight these two genes. An analysis of which gene families or functional categories are affected might be more informative (I would expect, as found in other species, that R genes and F box genes would be over-represented).

RESPONSE:

A total of 2,035 genes that contain large-effect SNPs were identified. Several genes were singled out in the text simply because they were well studied. We do not know if the mutations in these genes are associated with domestication or any specific traits. To avoid misunderstanding, corresponding sentences were removed from the revised manuscript.

Nevertheless, we believe that large-effect mutations are useful information for future studies of gene functions and breeding of lettuce. Instead of pointing out individual genes, we did GO and Pfam enrichment analysis of the affected genes. Genes encoding disease-related proteins and kinases were significantly overrepresented, which were consistent with previous findings^{17,18}. We also added the distribution of large-effect SNPs across groups/types in the updated Supplementary Table 2. Only eighteen cultivated lettuce accessions harbor the large-effect SNP in the LD gene, and all of the three *L. saligna* accessions harbor the large-effect SNP in the NPR3 gene. Such information is included in the supplementary data.

- About half of the genes with the strongest AEC patterns (including LHY) have no or almost no genes co-expressed in cultivated lettuce. That is the case for only one gene in wild lettuce. Is that expected? What is the reason for this pattern? Also, the conclusions the authors draw from the results presented in this section (line 257-258: “transcriptional networks became rewired in cultivated lettuce”) seem overly vague,

and should either be expanded or removed.

RESPONSE:

We reported this gene simply because this gene is well studied. It remains unknown whether the AEC (and the LHY in particular) is associated with domestication or any specific traits, and therefore we do not know its biological significance. We agree with the reviewer that the statement is vague and the sentence is removed as suggested. Removing the section of AEC analysis would have very minimal effect on the overall significance of our study.

- Did the authors find any evidence of introgression during domestication from other wild *Lactuca* species?

RESPONSE:

A comprehensive analysis was performed to identify introgression evidence. A total of 173 regions (71.88 Mb) were shown to be potentially introgressed from wild species using a likelihood ratio test method¹⁹. Our results are consistent with the fact that lettuce breeding is mainly based on the utilization of wild *Lactuca* germplasm from the primary gene pool (*L. serriola*)²⁰. The method and results were added to the revised manuscript.

- The results shown in Figure 2 should be further discussed; in particular, there is no comment in the text on the dispersion history of other horticultural types besides stem lettuce, or on the meaning of the migration rates (MIGS and MIGC) reported in Figure 2a.

RESPONSE:

In the previous version, stem lettuce was focused simply because it is more informative in term of evolution. Stem lettuce diverged from other types for thousands of years ago and was recorded in ancient books. In contrast, other horticultural types coexisted in the same region.

In the revised manuscript (in the section of “Demographic history inference for lettuce”), the dispersion history of other horticultural types was also discussed.

The meaning of the migration rates is explained in the revised manuscript. Migration Rate between Horticultural Types (MRHT) is used to replace MIGC, and Migration Rate between Wild and Cultivated lettuce (MRWC) was used to replace MIGS.

It should be noted that our final demographic model include five populations and there are 20 migration routes in the model. It will be over-parameterized our model if 20 migration parameters for all 20 migration routes. To simplify the model, only two symmetrical migration parameters were used for all the migration routes, one between different horticultural types (MRHT), and the other between wild and each of the cultivated types (MRWC).

Additionally, in the Discussion section, the authors twice use unpublished data to support their results (line 433 and 456). These data would be a great addition to the manuscript, as they would strengthen the authors’ conclusions. Otherwise, these

mentions should be avoided.

RESPONSE:

Centered QTL positions of non-shattering and absence of spines traits are added in the revised manuscript. The genetic mapping of four loci controlling leaf color in lettuce is also included in the manuscript.

The labels in several of the figures are too small and very hard to read even when magnified. Even in the original Word file, the resolution of Supplementary Figure 6 and 10 is too low for the image to be understood. Contrast should be increased for the PCA plots in figure 1b, since in several cases the individual data points are difficult to see.

RESPONSE:

We apologize for the low resolution of figures. The resolution in the old version was set low to minimize the file size. In the revised manuscript, we increased the resolution in these figures as well as figure 1b, Supplementary Figures 6 and 10.

Other points:

- Introduction: The focus on stem lettuce in China is a bit confusing at first, since the fact that stem lettuce is very different from other types and is predominantly cultivated in Asia is only mentioned in the discussion. That information should probably be moved to the introduction.

RESPONSE:

Thanks for the suggestion. The classification of different horticultural types is added to the “Introduction” section of the revised manuscript.

- line 90: it would be helpful to move here some of the information on what the 240 lines are and how they have been selected; most people wouldn't read the Material and Methods first and might be therefore confused with respect to what is what. Also, the main text never mentions the RILs and how they have been considered in the analyses.

RESPONSE:

Thanks for this great suggestion! We have moved the description of lettuce materials from the “Materials and Methods” section to the beginning of the “Results” section.

- Line 132-134: I am not sure what are the implications of this statement. While domesticated accessions are more abundant, they clearly have much less genetic variability than wild accessions.

RESPONSE:

In the old version of the manuscript, we implied that such deleterious mutations likely have much higher frequency in wild than in cultivars. However, we could not make such conclusion due to lack of statistic support. These two sentences have been removed from the revised manuscript.

- Line 254-256: Supplementary Figure 13: is the difference in average number of genes co-expressed with AEC genes in cultivated lettuce and *L. serriola* significant? There seem to be a lot more variation for that in domesticated lettuce (as mentioned before, almost half of the AEC genes have no or almost no gene co-expressed with them, and that is compensated by a smaller number of AEC genes with >1000 co-regulated genes). More in general, variation in the average number of co-regulated genes does not seem the most meaningful parameter to compare genetic networks.

RESPONSE:

As the reviewer pointed out that the comparison is not very meaningful, this paragraph has been removed from the revised manuscript. Instead, in the revised manuscript, we compared the coefficient of variation (CV) of gene expression in selected and non-selected genes during domestication. The CV of selected genes (51.72%) is significantly lower than that of the non-selected genes (71.26%). We conclude that the expression of the selected genes may have played an important role in domestication.

- Lines 367-427: I found this section really interesting, but all the connections between different loci are a bit hard to follow at times. If at all possible, it would be useful to condense all these information in a figure or table (they are currently scattered over several figures in the main text and supplementary material). I also found the expression data for the five candidate loci to be interesting results, and they could probably be moved to the main text

RESPONSE:

Thanks for the suggestion. We have combined Figure 3 and Figure 4 into one Figure (Figure 5a-c in the revised manuscript) and the expression data of the six candidate genes between red and green accessions were moved from the supplementary file to the Figure 5d. We also combined Manhattan plots from the eQTLs of six candidate genes into one Figure to do the comparison in the updated Supplementary Figure 17.

- Line 373: I guess that is not meant to be $\alpha = 1$ (the same value is reported also in line 684)

RESPONSE:

In our study, we used $\alpha = 0.05$ for eQTL mapping, and used $\alpha = 1$ for GWA of leaf color. These two Bonferroni-corrected thresholds have been widely used to control genome-wide type I error rate in GWA studies. For example, Chen *et al.*, Si *et al.* and Gao *et al.* used $\alpha = 0.05$ as the threshold⁸⁻¹⁰; Li *et al.*, Liu *et al.*, Wen *et al.*, and Yang *et al.* used $\alpha = 1$ as the threshold¹¹⁻¹⁴. Several studies^{15,16} used $\alpha = 1$ as the suggestive thresholds and $\alpha = 0.05$ as significant thresholds.

For GWA of leaf color, a signal was detected for locus 3 on chromosome 4 when using $\alpha = 1$, which was subsequently confirmed genetically. However, using $\alpha = 0.05$ failed to identify this locus, generating a false negative result.

To be consistent, both $\alpha = 0.05$ (significant threshold) and $\alpha = 1$ (suggestive threshold) in Bonferroni-correction was used in the GWAS of leaf color in the revised

manuscript.

- Line 373: according to Table 3, it should read “(Locus number 1 to 5)

RESPONSE:

Thanks for pointing out this typo. The change has been made in the revised manuscript.

- Line 468: These examples are not really meaningful and should be removed; bHLH and WD40 are widespread protein motifs, and MYB transcription factors regulate many other processes besides flavonoid biosynthesis.

RESPONSE:

The sentence has been removed from the revised manuscript.

- Line 863: it should read “Supplementary Table 9”.

RESPONSE:

Thanks for pointing out this typo. The change has been made in the revised manuscript.

- In Figure 1a, one individual of *L. serriola* appears to be much more similar than the others to intermediate and cultivated lettuce. Is that simply a mis-identified hybrid plant, or could that be (close to) the original population from where lettuce was domesticated?

RESPONSE:

The accession of individual that the reviewer pointed out is W44 (CGN04799). According to the Centre for Genetic Resources, the Netherlands (CGN) website (<http://www.wageningenur.nl/>), this individual was collected at 20 km south of Al-Fallujah in Iraq. This place is located in the Fertile Crescent where the lettuce was domesticated. Like most wild accessions, accession W44 has lobed leaf, flowering late, shattering (see picture attached below). Furthermore, W44 is phylogenetically closer to other wild accessions than the intermediate group (hybrids between wild and cultivars). We proposed that the close relationship between the wild accession W44 and cultivated lettuce is due to the fact that it is located near the center of domestication. Of course, more studies are required to support this hypothesis and rule out the possibility of gene flow from cultivars to the ancestor of W44. Above information is included in the Discussion of the revised manuscript.

The W44 plant

The location of W44

The green layer indicates the region of Fertile Crescent and the red marker suggests where the W44 was collected.

- Most of the main information in the Manhattan plot in Figure 4 is already presented in Figure 3a. It would make sense to fully combine the Manhattan plots in figures 3 and 4, to allow direct comparison (and move panel 4b to the Supplementary information).

RESPONSE:

Thanks for the suggestion. We agree with the suggestion and have combined Figure 3 and Figure 4 into one Figure (Figure 5) in the revised manuscript. And we moved Figure 4b to the Supplementary Figure 16 in the revised manuscript.

References

1. Vos, P.G. *et al.* Evaluation of LD decay and various LD-decay estimators in simulated and SNP-array data of tetraploid potato. *Theor Appl Genet* **130**, 123-135 (2017).
2. Huang, X. *et al.* Genome-wide association studies of 14 agronomic traits in rice landraces. *Nat Genet* **42**, 961-7 (2010).
3. Zhou, Z. *et al.* Resequencing 302 wild and cultivated accessions identifies genes related to domestication and improvement in soybean. *Nat Biotechnol* **33**, 408-14 (2015).
4. Mace, E.S. *et al.* Whole-genome sequencing reveals untapped genetic potential in Africa's indigenous cereal crop sorghum. *Nat Commun* **4**, 2320 (2013).
5. Xu, X. *et al.* Resequencing 50 accessions of cultivated and wild rice yields markers for identifying agronomically important genes. *Nat Biotechnol* **30**, 105-11 (2012).
6. Meyer, R.S. *et al.* Domestication history and geographical adaptation inferred from a SNP map of African rice. *Nat Genet* (2016).
7. Breseghello, F. & Sorrells, M.E. Association mapping of kernel size and milling quality in wheat (*Triticum aestivum* L.) cultivars. *Genetics* **172**, 1165-77 (2006).
8. Chen, W. *et al.* Genome-wide association analyses provide genetic and biochemical insights into natural variation in rice metabolism. *Nat Genet* **46**, 714-21 (2014).
9. Si, L. *et al.* OsSPL13 controls grain size in cultivated rice. *Nat Genet* (2016).
10. Gao, L., Zhao, G., Huang, D. & Jia, J. Candidate loci involved in domestication and improvement detected by a published 90K wheat SNP array. *Sci Rep* **7**, 44530 (2017).
11. Li, H. *et al.* Genome-wide association study dissects the genetic architecture of oil biosynthesis in maize kernels. *Nat Genet* **45**, 43-50 (2013).
12. Liu, Y. *et al.* Genome-wide association study of 29 morphological traits in *Aegilops tauschii*. *Sci Rep* **5**, 15562 (2015).
13. Wen, W. *et al.* Metabolome-based genome-wide association study of maize kernel leads to novel biochemical insights. *Nat Commun* **5**, 3438 (2014).
14. Yang, N. *et al.* Genome wide association studies using a new nonparametric model reveal the genetic architecture of 17 agronomic traits in an enlarged maize association panel. *PLoS Genet* **10**, e1004573 (2014).
15. Yang, W. *et al.* Combining high-throughput phenotyping and genome-wide association studies to reveal natural genetic variation in rice. *Nat Commun* **5**, 5087 (2014).
16. Duggal, P., Gillanders, E.M., Holmes, T.N. & Bailey-Wilson, J.E. Establishing an adjusted p-value threshold to control the family-wide type 1 error in genome wide association studies. *BMC Genomics* **9**, 516 (2008).
17. Clark, R.M. *et al.* Common sequence polymorphisms shaping genetic diversity in *Arabidopsis thaliana*. *Science* **317**, 338-42 (2007).
18. Lai, J. *et al.* Genome-wide patterns of genetic variation among elite maize inbred lines. *Nat Genet* **42**, 1027-30 (2010).
19. McNally, K.L. *et al.* Genomewide SNP variation reveals relationships among landraces and modern varieties of rice. *Proc Natl Acad Sci U S A* **106**, 12273-8 (2009).
20. Lebeda, A. *et al.* Wild *Lactuca* species, their genetic diversity, resistance to diseases and pests, and exploitation in lettuce breeding. *European Journal of Plant Pathology* **138**, 597-640 (2014).

REVIEWERS' COMMENTS:

Reviewer #1 (Remarks to the Author):

The manuscript has been improved and I am satisfied with the revision.

Reviewer #2 (Remarks to the Author):

The authors replied satisfactorily to all of my comments. I have only a few of remarks, mostly on additional material that has been added since the last version.

- Lines 275-287: this analysis uses groups of selected and non-selected genes, but the selection analysis is done only in the following section. The order in which the analyses are presented should be changed to avoid this confusion.

- Lines 286-287: this sentence is quite vague, and should probably be made more specific. It seems to me the main take-home message is that the expression of genes that underwent a selective sweep is more tightly regulated, as you would expect if these genes are indeed controlling traits that are differentiating cultivated from wild lettuce.

- Lines 488-492: This association does not seem especially meaningful, since "selected regions" can be found across the whole genome (so there would be selected regions associated with any possible QTL; Fig. 3C). I would remove these sentences.

A few minor suggestions:

- Lines 70-71: the meaning of this sentence would be clearer as "However, despite the presence of large amounts of phenotypic variation, genetic diversity in lettuce is limited."

- Line 311: reference should be "21", not "20".

Responses to reviewers' comments:

Reviewer #1 (Remarks to the Author):

The manuscript has been improved and I am satisfied with the revision.

RESPONSE:

We appreciate the reviewer's comments on our manuscript.

Reviewer #2 (Remarks to the Author):

The authors replied satisfactorily to all of my comments. I have only a few of remarks, mostly on additional material that has been added since the last version.

RESPONSE:

We appreciate the reviewer's comments on our manuscript.

- Lines 275-287: this analysis uses groups of selected and non-selected genes, but the selection analysis is done only in the following section. The order in which the analyses are presented should be changed to avoid this confusion.

RESPONSE:

As the reviewer pointed out, there seems to have problem of order. However, we cannot change the order accordingly since such change affects the flow of the text. To avoid confusion, we added "see below" in the revised manuscript to remind readers that the analysis on "selection" was performed in following sections.

- Lines 286-287: this sentence is quite vague, and should probably be made more specific. It seems to me the main take-home message is that the expression of genes that underwent a selective sweep is more tightly regulated, as you would expect if these genes are indeed controlling traits that are differentiating cultivated from wild lettuce.

RESPONSE:

Yes, we meant that selected genes are tightly "regulated", and we found low expression variation for selected genes. Such reduced expression variation may be associated with domestication and differentiation of different horticultural types. Our study also suggest that the reduced expression variation of selected genes was caused by decreased *cis*-QTLs. To summarize, we add the following sentence in the revised manuscript: "Our results suggest that the low expression variation of selected genes is due to less variation of their *cis*-acting regulatory regions." To avoid confusion, the last sentence (*i.e.* Lines 286-287 in the previous version) is removed in the revised manuscript.

- Lines 488-492: This association does not seem especially meaningful, since "selected regions" can be found across the whole genome (so there would be selected

regions associated with any possible QTL; Fig. 3C). I would remove these sentences.

RESPONSE:

We agree with this suggestion, and the sentences are removed from the revised manuscript.

A few minor suggestions:

- Lines 70-71: the meaning of this sentence would be clearer as “However, despite the presence of large amounts of phenotypic variation, genetic diversity in lettuce is limited.”

RESPONSE:

We apologize that a misuse of “genetics” caused confusion. “genetics” is changed to “genetic studies” in the revised manuscript.

- Line 311: reference should be “21”, not “20”.

RESPONSE:

We thank the reviewer for pointing this out. The change has been made in the revised manuscript.